# Revisiting Referring Expression Comprehension Evaluation in the Era of Large Multimodal Models

## Abstract

Referring expression comprehension (REC) involves localizing a target instance based on a textual description. Recent advancements in REC have been driven by large multimodal models (LMMs) like CogVLM, which achieved 92.44% accuracy on RefCOCO. However, this study questions whether existing benchmarks such as RefCOCO, RefCOCO+, and RefCOCOg, capture LMMs' comprehensive capabilities. We begin with a manual examination of these benchmarks, revealing high labeling error rates: 14% in RefCOCO, 24% in RefCOCO+, and 5% in RefCOCOg, which undermines the authenticity of evaluations. We address this by excluding problematic instances and reevaluating several LMMs capable of handling the REC task, showing significant accuracy improvements, thus highlighting the impact of benchmark noise. In response, we introduce Ref-L4, a comprehensive REC benchmark, specifically designed to evaluate modern REC models. Ref-L4 is distinguished by four key features: 1) a substantial sample size with 45,341 annotations; 2) a diverse range of object categories with 365 distinct types and varying instance scales from 30 to 3,767; 3) lengthy referring expressions averaging 24.2 words; and 4) an extensive vocabulary comprising 22,813 unique words. We evaluate a total of 24 large models on Ref-L4 and provide valuable insights. The cleaned versions of RefCOCO, RefCOCO+, and RefCOCOg, as well as our Ref-L4 benchmark and evaluation code, will be made available to the community.

## 1 Introduction

Referring expression comprehension (REC) (Nagaraja et al., 2016; Wang et al., 2024d) involves the task of localizing a specific target instance based on a given textual description. The advancement of REC has been significantly propelled by the superior language processing capabilities of large language models (LLMs) (Touvron et al., 2023a;b; Meta, 2024). This progress is particularly evident in the exceptional performance of large multimodal models (LMMs) (He et al., 2024; Wang et al., 2024a; Zhao et al., 2024) on well-known benchmarks such as RefCOCO (Yu et al., 2016), RefCOCO+ (Yu et al., 2016), and RefCOCOg (Mao et al., 2016). These models have demonstrated remarkable accuracy, with CogVLM (Wang et al., 2023b), for instance, achieving an impressive accuracy rate of 92.44% on the RefCOCO benchmark.

This paper begins with a critical question: do existing REC benchmarks truly capture the comprehensive capabilities of LMMs? The foundational benchmarks, RefCOCO, RefCOCO+, and RefCOCOg, were introduced sequentially in 2015, 2016, and 2016, respectively. In RefCOCO, the referring expressions are notably succinct, ranging from single words like "*lady*" and "*yellow*" to brief descriptions such as "*far left person*" and "*white shirt*". RefCOCO+ intentionally excludes locational prepositions commonly found in RefCOCO, favoring short yet semantically rich expressions like "*plastic cup with just ice*" and "*man on screen*". Conversely, RefCOCOg provides more elaborate annotations, including examples such as "*a table of food, with plates, a pizza, pitchers, and glasses*" and "*a red and white checkered table with two wooden chairs*". These variations highlight the evolution and complexity of referring expressions across different benchmarks, raising the question of whether they can effectively assess the nuanced capabilities of modern LMMs in understanding diverse linguistic inputs and associating languages with visual elements.

Table 1: Statistics of the labeling error rates for RefCOCO, RefCOCO+, and RefCOCOg, respectively. For each benchmark, the statistics are conducted on the combination of the validation and test sets.

| Benchmark | Annotations | Errors | Labeling Error Rate |
|---|---|---|---|
| RefCOCO (Yu et al., 2016) | 21,586 | 3,054 | 14% |
| RefCOCO+ (Yu et al., 2016) | 21,373 | 5,201 | 24% |
| RefCOCOg (Mao et al., 2016) | 14,498 | 675 | 5% |

Table 2: The performance of four LMMs capable of handling the REC task on both the cleaned and original versions of the RefCOCO, RefCOCO+, and RefCOCOg benchmarks, using the conventional accuracy as the evaluation metric. The evaluation is performed on the combination of the validation and test sets for each benchmark. †: models fine-tuned on the specific dataset.

| Benchmark | ONE-PEACE† | OFA-L† | OFA-L | Qwen-VL | CogVLM-Grounding |
|---|---|---|---|---|---|
| RefCOCO | 92.15 | 89.85 | 85.13 | 88.51 | 92.44 |
| RefCOCO (Cleaned) | 94.11 **(+1.96)** | 92.06 **(+2.22)** | 87.95 **(+2.81)** | 90.68 **(+2.18)** | 94.58 **(+2.13)** |
| RefCOCO+ | 88.14 | 85.06 | 77.56 | 82.52 | 88.55 |
| RefCOCO+ (Cleaned) | 90.79 **(+2.66)** | 87.38 **(+2.32)** | 80.50 **(+2.94)** | 85.60 **(+3.08)** | 91.43 **(+2.87)** |
| RefCOCOg | 89.18 | 84.77 | 79.25 | 85.11 | 90.67 |
| RefCOCOg (Cleaned) | 90.75 **(+1.57)** | 86.39 **(+1.62)** | 80.89 **(+1.64)** | 86.79 **(+1.68)** | 92.36 **(+1.68)** |

Table 3: Comparison between our Ref-L4 benchmark and other REC benchmarks, including RefCOCO, RefCOCO+, and RefCOCOg. For the latter three benchmarks, we combine their validation and test sets for statistics. The instance size and image size are represented by their respective square roots. Avg. length: average length of annotations. Vocab.: vocabulary size.

| Benchmark | Images | Instances | Annotations | Categories | Avg. Length | Instance Size | Image Size | Vocab. |
|---|---|---|---|---|---|---|---|---|
| RefCOCO | 3,000 | 7,596 | 21,586 | 71 | 3.6 | 105 - 607 | 230 - 640 | 3,525 |
| RefCOCO+ | 3,000 | 7,578 | 21,373 | 71 | 3.6 | 105 - 607 | 230 - 640 | 4,387 |
| RefCOCOg | 3,900 | 7,596 | 14,498 | 78 | 8.4 | 83 - 610 | 277 - 640 | 5,050 |
| Ref-L4 (Ours) | 9,735 | 18,653 | 45,341 | 365 | 24.2 | 30 - 3,767 | 230 - 6,606 | 22,813 |

**Labeling Error Rates of Existing Benchmarks.** To begin, we manually assess the labeling error rates of the validation and test sets in RefCOCO, RefCOCO+, and RefCOCOg, discovering a high error rate across these benchmarks. The labeling errors include, typos, misalignment between referring expressions and target instances, as well as inaccurate bounding box annotations, as depicted in Section A. As illustrated in Table 1, the labeling error rates for RefCOCO, RefCOCO+, and RefCOCOg are 14%, 24%, and 5%, respectively, indicating that evaluations performed on these benchmarks may lack authenticity.

**Reevaluation on RefCOCO, RefCOCO+ and RefCOCOg.** In response, we manually exclude the problematic instances from the validation and test sets of RefCOCO, RefCOCO+, and RefCOCOg. Subsequently, we reevaluate four LMMs capable of handling the REC task—namely ONE-PEACE (Wang et al., 2023a), OFA-L (Wang et al., 2022), Qwen-VL (Bai et al., 2023), and CogVLM-Grounding (Wang et al., 2023b)—on both the cleaned and original versions of these datasets, as shown in Table 2. Across all models and cleaned benchmarks, we observe a significant accuracy improvement, ranging from 1.57 to 3.08, compared to their performance on the original versions. This demonstrates that noise in the benchmarks has impacted the models' true capabilities. *To support further research in the REC field, we will release the cleaned versions of RefCOCO, RefCOCO+, and RefCOCOg.*

**Ref-L4: A Comprehensive REC Benchmark for Modern LMM Evaluation.** We present Ref-L4, where L4 signifies four key aspects: a Large number of testing samples, Large diversity in object categories and instance scales, Long referring expressions, and a Large vocabulary. These fea-

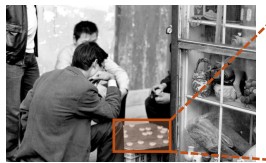

The pale green rectangular eraser features a depiction of a bear, accompanied by the word "ERASER" inscribed in green. A transparent plastic covering with patterns partially envelops it. Positioned at the bottom right corner of the picture, the eraser rests on a cluttered desk surrounded by an assortment of artistic materials and drawings.

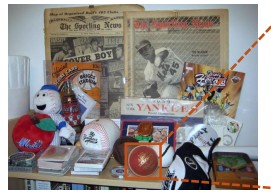

The game board is a square, wooden framework positioned at the lower part of the picture, featuring a grid of tiny recessed circles containing circular tokens. It is placed on the floor close to a shelf showcasing an assortment of objects, such as a teapot and bottles. The elevated borders of the board indicate that it is intended for gameplay, potentially involving tactics or positioning.

A decorative baseball with a unique red and gold color scheme, situated amongst various baseball memorabilia.

Figure 1: Examples from our Ref-L4 benchmark. We offer a detailed referring expression for each target instance represented by a bounding box. Zoom in for better visualization.

tures make Ref-L4 a comprehensive benchmark for assessing the REC capabilities of contemporary LMMs. Table 3 provides a detailed comparison between Ref-L4 and other benchmarks including RefCOCO, RefCOCO+, and RefCOCOg. Our Ref-L4 benchmark stands out due to the following characteristics:

- *Large-Scale.* Ref-L4 includes 9,735 images, 18,653 unique instances, and a total of 45,341 annotations, significantly surpassing RefCOCO, RefCOCO+, and RefCOCOg. For instance, RefCOCOg offers 3,900 images, 7,596 instances, and 14,498 annotations.

- *High Diversity.* Ref-L4 features 365 unique categories. Since the RefCOCO series derive from the COCO 2014 dataset, they encompass up to 78 categories. Additionally, our benchmark covers a wider range of instance scales, from 30 to 3,767, measured by the square root of the instance area.

- *Lengthy Referring Expressions.* Each referring expression in Ref-L4 is a detailed description of a specific instance, with lengths ranging from 33 to 117 words and an average of 24.2 words. In comparison, the average annotation lengths in RefCOCO, RefCOCO+, and RefCOCOg are 3.6, 3.6, and 8.4 words, respectively. Examples can be found in Figure 1.

- *Extensive Vocabulary.* Due to the detailed nature of the referring expressions, Ref-L4 boasts a large vocabulary of 22,813 words, which is four to six times larger than those of Ref-COCO, RefCOCO+, and RefCOCOg.

**Evaluation on Ref-L4.** We conduct an evaluation of 24 representative LMMs that can perform the REC task. In addition to the standard accuracy metric, which considers predictions with an IoU greater than 0.5 as accurate ($Acc_{0.5}$), we also report accuracies at higher IoU thresholds: $Acc_{0.75}$ and $Acc_{0.9}$. Furthermore, we introduce a mean accuracy (mAcc), calculated as the average accuracy from $Acc_{0.5}$ to $Acc_{0.95}$ in increments of 0.05. To gain deeper insights into the models' capabilities, we conduct a detailed analysis of REC performance across different instance scales and categories. *The Ref-L4 benchmark and the evaluation code will be made available to the community.*

## 2 RELATED WORK

**REC and Its Benchmarks.** Referring Expression Comprehension (REC) (Qiao et al., 2020; Nagaraja et al., 2016; Zheng et al., 2022; Kazemzadeh et al., 2014; Pi et al., 2023; Zhang et al., 2019)

is a task that involves identifying a specific object within an image based on a given referring expression. Unlike object detection (Lin et al., 2014; Krishna et al., 2017; Shao et al., 2019; Redmon et al., 2016; Carion et al., 2020), which operates within fixed categories and a single visual modality, REC necessitates understanding free-form text to locate objects of any category. Phrase Grounding (Plummer et al., 2015; Wu et al., 2020; Gupta et al., 2019; Liu et al., 2023; Li et al., 2022; Zhang et al., 2022; Wang et al., 2020b) is similar but typically involves shorter phrases and identifies multiple regions, whereas REC requires parsing longer expressions to pinpoint a single unique region. This complexity makes REC an ideal task for evaluating emerging large multimodal models. Current REC benchmarks such as RefCOCO (Yu et al., 2016), RefCOCO+(Yu et al., 2016), and RefCOCOg(Mao et al., 2016) include tens of thousands of annotations but are limited by their short expression lengths—averaging 3.6, 3.6, and 8.4 words, respectively. Additionally, they encompass fewer than 80 categories, lacking real-world diversity. Other REC benchmarks (Liu et al., 2019; Chen et al., 2023c; Qiu et al., 2022; Chen et al., 2020; Wang et al., 2024c; Kurita et al., 2023; Wang et al., 2020a; Cirik et al., 2020; Bu et al., 2022; Gao et al., 2023; De Vries et al., 2017; Jia et al., 2024) are often designed for specific scenarios. For example, CLEVR-Ref+(Liu et al., 2019) focuses on simple objects like boxes, spheres, and cylinders. SK-VG(Chen et al., 2023c) integrates prior scene knowledge as additional input, while RefCrowd (Qiu et al., 2022) targets identifying a person within a crowd. By contrast, we introduce Ref-L4, a more general and comprehensive benchmark encompassing 365 categories and 45,341 annotations. Ref-L4 features expressions averaging 24.2 words and a vocabulary of 22,813 words, facilitating the accurate evaluation of REC models on complex expressions and diverse objects.

**REC Models.** The evolution of REC models has transitioned from specialized models (Kamath et al., 2021; Yu et al., 2018; Liu et al., 2017; Su et al., 2019; Zheng et al., 2019; Yan et al., 2023; Zou et al., 2024) to generalist models or large multimodal models (LMMs)(Wang et al., 2023b; Lin et al., 2023; Gao et al., 2024; Wang et al., 2023a; Bai et al., 2023; Chen et al., 2023a; Wei et al., 2023; Zhang et al., 2024a; Zhan et al., 2023; 2024; Pramanick et al., 2023; Zhang et al., 2023; Wang et al., 2024b; Shen et al., 2024; Ma et al., 2024; Qi et al., 2024; KOSAREVA, 2024). Notable examples of these LMMs include CogVLM-Grounding(Wang et al., 2023b), SPHINX (Lin et al., 2023; Gao et al., 2024), ONE-PEACE (Wang et al., 2023a), Qwen-VL-Chat (Bai et al., 2023), MiniGPTv2 (Chen et al., 2023a), and Lenna (Wei et al., 2023). These models, benefiting from larger model sizes and extensive training on diverse datasets, exhibit remarkable performance on conventional REC datasets. For example, CogVLM-Grounding achieves an accuracy of $94.58\%$ on RefCOCO (cleaned). Additionally, the performance gap among models is shrinking, with many LMMs surpassing $90\%$ accuracy. This performance saturation raises concerns about the adequacy of current REC benchmarks for making meaningful comparisons. In response, we propose Ref-L4, a more comprehensive and challenging benchmark. We have also conducted rigorous evaluations of 24 LMM models, offering holistic comparisons that highlight their weaknesses and suggest directions for improvement.

## 3 REF-L4

### 3.1 BENCHMARK CREATION

**Data Sources.** Our benchmark is derived from two sources: 1) our cleaned validation and test sets of the RefCOCO (Yu et al., 2016), RefCOCO+ (Yu et al., 2016), and RefCOCOg (Mao et al., 2016) datasets; and 2) the test set from the large-scale object detection dataset Objects365 (Shao et al., 2019). The Objects365 dataset provides a broader range of categories, varying instance sizes, higher image resolutions, and more intricate scenes. In the RefCOCO series, each instance includes a bounding box, a category name, and an extremely brief expression like "right teddy bea". In contrast, the Objects365 benchmark labels each instance with mainly a bounding box and the relevant category.

For the RefCOCO (cleaned) series, we begin by consolidating duplicate images and instances, resulting in a subset of $6,502$ images containing $14,186$ unique instances. For Objects365, we select samples from its testing set based on several criteria: 1) each image has both height and width greater than 800 pixels; 2) each image is sufficiently complex, containing more than 10 categories and 20 instances; 3) each instance has a square normalized size $\sqrt{(hw)/(HW)}$ greater than 0.05, where $(h, w)$ represents the instance size and $(H, W)$ denotes the image size; 4) we randomly sample $N$ instances for each of the 365 classes defined in Objects365, with

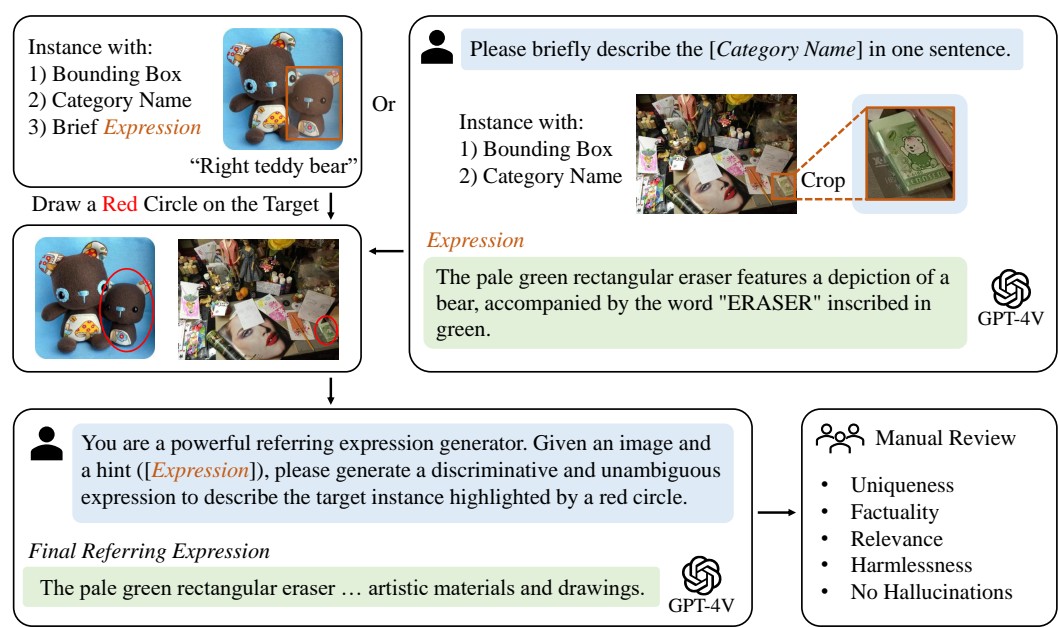

Figure 2: Pipeline of generating a referring expression for a target instance.

$N = \min(35,$ the number of instances for the specific class); 5) we review and exclude instances with erroneous bounding box annotations or those difficult to describe uniquely. For a few rare classes, we relax criterion-1 to 512 pixels and criterion-2 to 10 instances. Consequently, we collect $3,233$ images and $4,467$ instances from Objects365. Overall, our Ref-L4 benchmark comprises $9,735$ images and $18,653$ instances, sourced from the RefCOCO series and Objects365.

**Referring Expression Generation.** Given a target instance and its corresponding image, we leverage GPT-4V with human reviewers in the loop to generate its precise and detailed referring expressions. Figure 2 illustrates the three-step generation process:

*Step-1*: Each instance in the Objects365 dataset is linked to a bounding box and a *category name*. We begin by cropping these instances from the original images. Next, we input each cropped area along with the prompt detailed in Section B.1 into GPT-4V to produce a context-independent description. For instances from the RefCOCO series, this step is omitted as each instance already has a brief expression.

*Step-2*: Drawing inspiration from recent studies on GPT-4V (Yang et al., 2023), where GPT-4V is able to pay more attention to instances highlighted by a red circle within an image, we similarly encircle the target instance in red to facilitate GPT-4V in generating a context-aware referring expression. Following this, as depicted in Figure 2, we process the image and use the prompt outlined in Section B.2 to generate a context-aware referring expression for each instance. We instruct GPT-4V to describe various features such as color, size, position, and context. Additionally, we provide a hint (the context-independent description from Step-1) in the prompt to mitigate hallucination issues, resulting in more accurate descriptions.

*Step-3*: We manually review all generated referring expressions to correct any hallucination issues. We ensure that each expression uniquely describes the instance and is factual, accurate, and harmless.

**Annotation Expansion.** To date, we have compiled 18,653 unique referring expressions, each describing a distinct instance. To assess the robustness of REC models to diverse language inputs, we employ a two-stage rephrasing process to expand our benchmark: 1) utilizing GPT-4 with the prompt detailed in Section B.3, to generate rephrased versions of each expression; 2) conducting a manual review to ensure that the rephrased expressions are unique, factual, relevant, and harmless. Consequently, our final Ref-L4 benchmark encompasses 9,735 images with 45,341 referring expressions, each accurately describing one of the 18,653 unique instances.

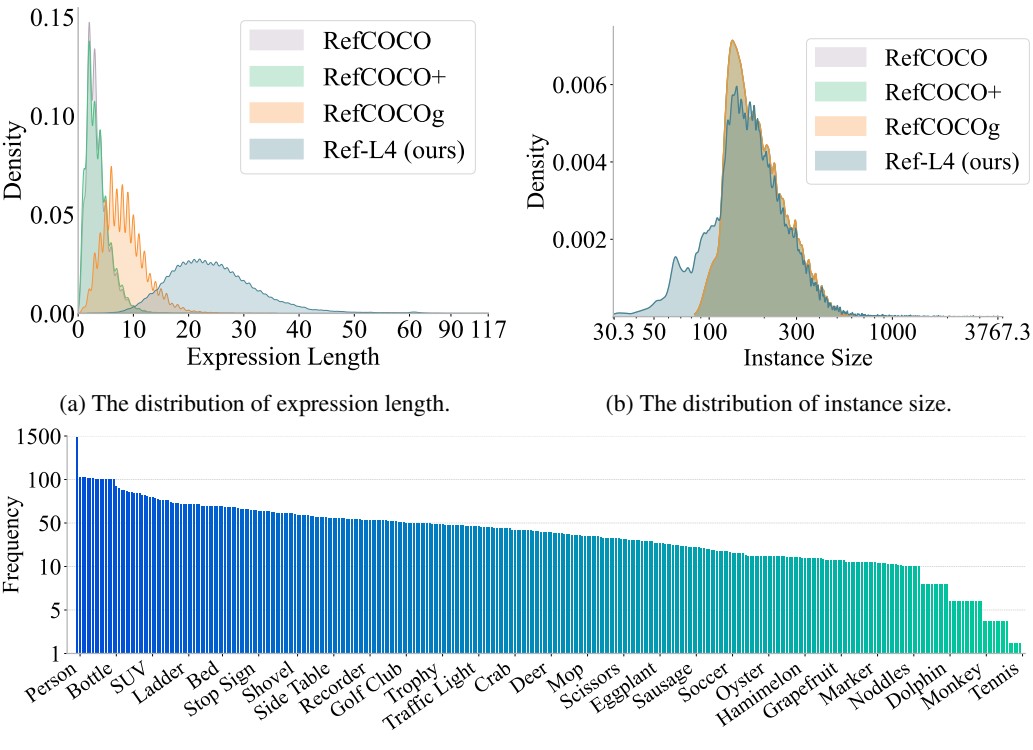

(a) The distribution of expression length.

(b) The distribution of instance size.

(c) The distribution of instance numbers over 365 categories.

Figure 3: Analysis of referring expression length, instance size, and category distribution.

## 3.2 ANALYSIS

**Expression Length.** Figure 3a illustrates the distribution of expression lengths across four different datasets: RefCOCO, RefCOCO+, RefCOCOg, and our Ref-L4. Due the the high overlap of data samples, RefCOCO and RefCOCO+ exhibit similar distributions, with a high density of shorter expressions peaking at around 3.6 words. RefCOCOg features slightly longer expressions on average, peaking at approximately 8.4 words. In contrast, our Ref-L4 displays a significantly different distribution, with expressions ranging much longer, peaking at around 24.2 words and having a long tail extending up to 117 words. This suggests that our Ref-L4 benchmark is designed to push the boundaries of current REC models, requiring them to process and comprehend more intricate and detailed descriptions.

**Instance Size.** In Figure 3b, we present a density plot comparing the instance sizes across four benchmarks. We define the instance size as the square root of the normalized size, $\sqrt{(hw)/(HW)}$, where $(h, w)$ represents the dimensions of the instance and $(H, W)$ represents the dimensions of the image. All benchmarks exhibit a peak density around an instance size of 160. Our Ref-L4 benchmark shows a wider distribution range compared to the other three, indicating that our Ref-L4 captures a broader spectrum of instance sizes.

**Categories.** Our Ref-4L benchmark comprises 18,653 instances spanning 365 distinct categories, providing more complex and diverse evaluation scenarios. In contrast, RefCOCO and RefCOCO+ consists of 71 categories, while RefCOCOg covers 78 categories. Figure 3c presents the distribution of instances among these 365 categories. Notably, the ten categories with the highest number of instances are "Person", "Chair", "Hat", "Desk", "Lamp", "Cabinet/shelf", "Car", "Sneakers", "Handbag/Satchel", and "Flag".

**Scenes.** We provide a detailed scene analysis on our benchmark. We start by referencing the 365 scene categories from the Places365 benchmark (Zhou et al., 2017), known for being the most extensive dataset in scene recognition. These 365 categories are then consolidated into 20 broader groups using GPT-4o. Each image in our benchmark is processed by GPT-4o to predict its corresponding scene category, with manual corrections applied to ensure accuracy. The resulting statistics on scene

Table 4: Scene diversity across 20 consolidated categories, predicted by GPT-4o and manually corrected, based on the combined validation and test sets.

| Category | Percentage(%) | Category | Percentage(%) |
|---|---|---|---|
| Residential & Domestic Spaces | 19.68 | Entertainment | 2.88 |
| Catering & Dining | 16.36 | Recreational Facilities | 2.46 |
| Urban Scenes & Streetscapes | 9.14 | Water & Maritime Scenes | 2.43 |
| Transportation & Transit | 8.89 | Industrial & Workplaces | 1.92 |
| Sports & Exercise | 8.71 | Outdoor & Adventure | 1.75 |
| Wildlife | 6.25 | Hospitality, Resorts & Lodging | 1.28 |
| Commercial & Retail Spaces | 5.18 | Infrastructure & Public Services | 1.03 |
| Educational & Cultural Facilities | 4.42 | Health & Care Facilities | 0.51 |
| Agriculture & Rural | 3.79 | Natural Landscapes | 0.11 |
| Parks & Outdoor Leisure | 3.16 | Scientific Interest | 0.05 |

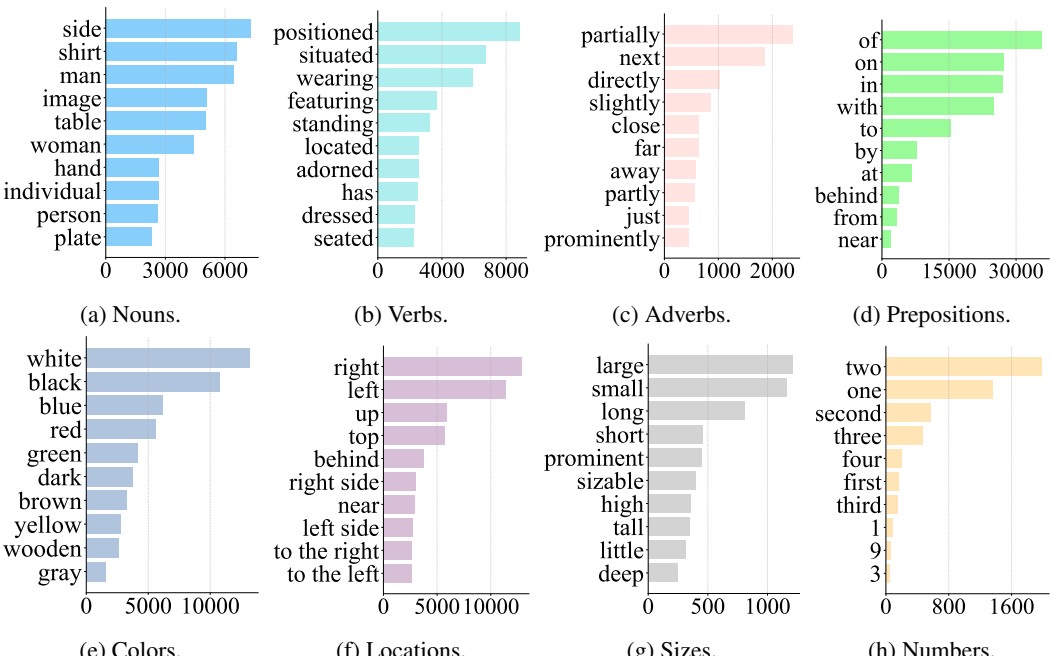

(a) Nouns.  (b) Verbs.  (c) Adverbs.  (d) Prepositions.

(e) Colors.  (f) Locations.  (g) Sizes.  (h) Numbers.

Figure 4: The frequency of the 10 most frequently used words in each part-of-speech category, as parsed using the SpaCy library.

diversity are summarized in the Table 4, with the combined validation and test sets used for this analysis.

**Vocabulary.** Our benchmark's referring expressions comprise a vocabulary totaling 22,813 unique words. This is significantly larger than the vocabulary sizes of RefCOCO, RefCOCO+, and RefCOCOg, which are 3,525, 4,387, and 5,050 words, respectively. Figure 4 illustrates the 10 most frequently used nouns, verbs, adverbs, and prepositions, along with nouns indicating colors, locations, sizes, and numbers across all annotations.

### 3.3 EVALUATION

**Evaluation Metrics.** We propose three distinct evaluation protocols:

1. *Accuracy.* This is the conventional metric used in REC. For a given referring expression and corresponding image, the target instance is considered successfully localized if the IoU between the predicted bounding box and the ground truth exceeds 0.5. Accuracy is then calculated as the ratio of successfully localized samples to the total number of samples, referred to as $Acc_{0.5}$ in this work. To better assess the localization capabilities of modern REC models, we also report

Table 5: Performance evaluation across 24 models on our Ref-L4 benchmark. NVIDIA A100 GPUs (80G) are utilized. The symbol † denotes models that outputs segmentation masks.

| Model | Val+Test | | | | Val | Test |
|---|---|---|---|---|---|---|
| | $Acc_{0.5}$ | $Acc_{0.75}$ | $Acc_{0.9}$ | mAcc | mAcc | mAcc |
| GPT-4V (OpenAI, 2023a;b;c) | 9.91 | 1.19 | 0.12 | 2.88 | 2.96 | 2.85 |
| KOSMOS-2 (Peng et al., 2023) | 48.53 | 38.34 | 17.54 | 34.72 | 34.89 | 34.64 |
| OFA-Tiny (Wang et al., 2022) | 55.21 | 43.22 | 27.70 | 41.44 | 41.53 | 41.40 |
| OFA-Large (Wang et al., 2022) | 72.53 | 62.31 | 45.02 | 59.17 | 59.42 | 59.07 |
| Ferret-7b (You et al., 2023) | 57.54 | 42.44 | 21.01 | 40.29 | 40.31 | 40.28 |
| Ferret-13b (You et al., 2023) | 64.44 | 49.04 | 27.46 | 46.88 | 47.31 | 46.71 |
| GroundingGPT (Li et al., 2024) | 60.84 | 40.48 | 12.00 | 38.19 | 38.42 | 38.09 |
| Shikra-7b (Chen et al., 2023b) | 65.06 | 39.62 | 10.45 | 38.60 | 38.91 | 38.47 |
| Lenna (Wei et al., 2023) | 65.90 | 58.55 | 45.58 | 55.69 | 55.88 | 55.60 |
| MiniGPTv2 (Chen et al., 2023a) | 66.93 | 50.50 | 25.30 | 47.15 | 47.43 | 47.03 |
| Qwen-VL-Chat (Bai et al., 2023) | 73.80 | 58.05 | 37.16 | 55.94 | 56.18 | 55.83 |
| ONE-PEACE (Wang et al., 2023a) | 70.82 | 60.09 | 36.12 | 55.07 | 55.49 | 54.89 |
| SPHINX-MoE (Gao et al., 2024) | 66.23 | 44.90 | 15.32 | 42.38 | 42.80 | 42.21 |
| SPHINX-MoE-1k (Gao et al., 2024) | 74.45 | 62.70 | 38.85 | 58.07 | 58.35 | 57.95 |
| SPHINX (Lin et al., 2023) | 74.78 | 53.65 | 21.15 | 50.09 | 50.33 | 49.99 |
| SPHINX-1k (Lin et al., 2023) | 78.52 | 62.17 | 32.95 | 57.57 | 57.91 | 57.42 |
| SPHINX-v2-1k (Lin et al., 2023) | 81.31 | 70.49 | 46.59 | 65.39 | 65.67 | 65.27 |
| CogVLM-Grounding (Wang et al., 2023b) | **81.70** | **70.77** | **48.35** | **66.09** | **66.25** | **66.02** |
| PixelLM-7B† (Ren et al., 2023) | 41.83 | 27.57 | 13.32 | 27.10 | 27.09 | 27.11 |
| PixelLM-13B† (Ren et al., 2023) | 49.89 | 35.37 | 18.42 | 34.10 | 34.52 | 33.92 |
| LISA-Explanatory† (Lai et al., 2023) | 65.12 | 52.35 | 38.26 | 50.77 | 50.89 | 50.72 |
| LISA† (Lai et al., 2023) | 66.23 | 54.02 | 39.73 | 52.18 | 52.44 | 52.07 |
| PSALM† (Zhang et al., 2024b) | 67.26 | 58.22 | 44.11 | 55.46 | 55.68 | 55.37 |
| GlaMM† (Rasheed et al., 2023) | **71.90** | **60.27** | **45.15** | **57.89** | **58.16** | **57.78** |

accuracies at higher IoU thresholds: $Acc_{0.75}$, $Acc_{0.9}$, and mAcc, which is the average accuracy from $Acc_{0.5}$ to $Acc_{0.95}$ in increments of 0.05.

2. *Scale-Aware Performance.* To gain deeper insights into model capabilities, we report performance based on instance sizes: small, medium, and large. The size of an instance is defined as the square root of its area, $\sqrt{(hw)}$, where $(h, w)$ are the dimensions of the instance. Small instances are those with a size less than 128, medium instances are between 128 and 256, and large instances exceed 256. In total, there are 9345, 23280, and 12716 referring expressions describing 2,954 small, 10,442 medium, and 5,257 large instances, respectively.

3. *Per-Category Performance.* Our benchmark encompasses a wide range of categories, up to 365 in total. We provide an evaluation protocol to assess performance on a per-category basis.

**Benchmark Division.** Modern large multimodal models (LMMs) that are able to handle the REC task typically use unrestricted and extensive data for training. Our Ref-L4 benchmark is designed to assess the capabilities of these advanced models without imposing any limitations on the training data sources. The benchmark is divided into two subsets: a validation set, comprising 30% of the data with 7,231 images, 10,311 instances, and 13,420 referring expressions; and a test set, comprising 70% of the data with 9,467 images, 17,242 instances, and 31,921 referring expressions. Given that our benchmark includes instances from 365 categories, we ensure that each category has at least one sample in both the validation and test sets. While we provide these two splits, we encourage the combined use of both sets for model evaluation, especially in the current LMM era, where the use of unrestricted training data is prevalent.

## 4 EXPERIMENTS

**Main Result.** We evaluate a total of 24 LMMs that can perform the REC task, dividing them into two categories based on their output type: those that produce bounding boxes and those that produce segmentation masks. For models that output segmentation masks, we convert these masks into tight bounding boxes to enable evaluation on our Ref-L4 benchmark. Table 5 presents the performance of these models on the validation set, test set, and the combined set, using the metrics defined in

Table 6: Scale-aware evaluation across 24 models on our Ref-L4 benchmark.

| Model | Small Size | | Medium Size | | Large Size | |
|---|---|---|---|---|---|---|
| | $Acc_{0.5}$ | mAcc | $Acc_{0.5}$ | mAcc | $Acc_{0.5}$ | mAcc |
| GPT-4V (OpenAI, 2023a;b;c) | 2.13 | 0.49 | 10.29 | 2.78 | 14.93 | 4.83 |
| KOSMOS-2 (Peng et al., 2023) | 24.19 | 11.63 | 46.95 | 32.91 | 69.32 | 54.98 |
| OFA-Tiny (Wang et al., 2022) | 17.91 | 11.49 | 65.13 | 49.00 | 64.46 | 49.61 |
| OFA-Large (Wang et al., 2022) | 40.13 | 27.07 | 81.03 | 66.49 | 80.78 | 69.36 |
| Ferret-7b (You et al., 2023) | 30.93 | 14.57 | 62.40 | 43.72 | 68.18 | 52.92 |
| Ferret-13b (You et al., 2023) | 36.46 | 17.88 | 70.50 | 51.86 | 73.92 | 59.09 |
| GroundingGPT (Li et al., 2024) | 24.43 | 10.28 | 67.67 | 41.04 | 75.09 | 53.47 |
| Shikra-7b (Chen et al., 2023b) | 43.91 | 18.50 | 75.98 | 46.27 | 60.60 | 39.34 |
| Lenna (Wei et al., 2023) | 31.02 | 23.48 | 72.90 | 61.53 | 78.72 | 68.66 |
| MiniGPTv2 (Chen et al., 2023a) | 32.99 | 14.85 | 73.67 | 51.16 | 79.52 | 63.53 |
| Qwen-VL-Chat (Bai et al., 2023) | 47.66 | 26.26 | 79.80 | 61.06 | 82.01 | 68.37 |
| ONE-PEACE (Wang et al., 2023a) | 22.18 | 13.98 | 83.26 | 63.39 | 83.81 | 70.04 |
| SPHINX-MoE (Gao et al., 2024) | 39.48 | 16.39 | 72.97 | 46.38 | 73.55 | 54.17 |
| SPHINX-MoE-1k (Gao et al., 2024) | 58.96 | 37.61 | 77.80 | 61.53 | 79.70 | 66.77 |
| SPHINX (Lin et al., 2023) | 48.82 | 22.08 | 80.56 | 54.10 | 83.27 | 63.34 |
| SPHINX-1k (Lin et al., 2023) | 59.48 | 33.21 | 82.95 | 61.82 | 84.40 | 67.68 |
| SPHINX-v2-1k (Lin et al., 2023) | 65.23 | 43.43 | 84.00 | 68.45 | **88.21** | **75.91** |
| CogVLM-Grounding (Wang et al., 2023b) | **75.06** | **52.85** | **86.43** | **71.31** | 77.91 | 66.25 |
| PixelLM-7B[†] (Ren et al., 2023) | 8.25 | 4.05 | 43.90 | 27.33 | 62.72 | 43.64 |
| PixelLM-13B[†] (Ren et al., 2023) | 17.05 | 8.54 | 53.40 | 35.48 | 67.59 | 50.34 |
| LISA-Explanatory[†] (Lai et al., 2023) | 39.11 | 27.16 | 70.03 | 54.61 | 75.25 | 61.09 |
| LISA[†] (Lai et al., 2023) | 39.24 | 27.49 | 71.17 | 56.05 | 77.01 | 63.22 |
| PSALM[†] (Zhang et al., 2024b) | 37.35 | 28.43 | 75.06 | 61.79 | 74.97 | 63.74 |
| GlaMM[†] (Rasheed et al., 2023) | **47.07** | **34.36** | **77.17** | **62.28** | **80.50** | **67.14** |

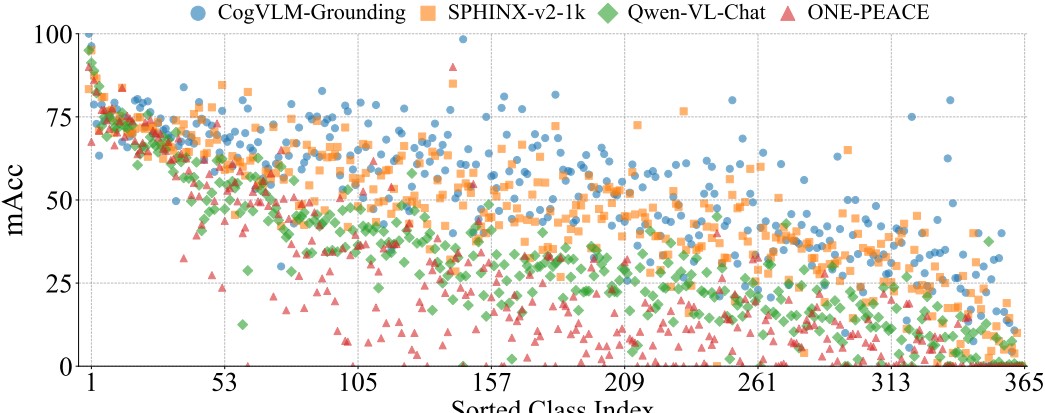

Figure 5: Category-wise performance of the four top-performing models on the val+test set, sorted in descending order based on their average per-category performance. The performance of all models can be found in Section C.1.

Section 3.3. The evaluation prompt of GPT-4V is available in Section B.4. Among the models that output bounding boxes, CogVLM-Grounding (Wang et al., 2023b) shows the best performance, while GlaMM (Rasheed et al., 2023) leads in performance among the models that output masks.

**Category-Wise Performance.** Each instance in our benchmark is assigned a category label from one of 365 classes. Figure 5 illustrates the performance of the top four models across these categories, sorted in descending order based on their average per-category performance. The results indicate a training bias issue, as all four models exhibit poor performance on some common categories.

**Scale-Aware Evaluation.** In Section 3.3, we present a scale-aware evaluation to assess the model's ability to handle different instance scales. Specifically, we categorize all samples in our benchmark into three sets based on instance size: small, medium, and large. The performance of 24 models is

Table 7: Evaluation of four models on the RES benchmark, extended from our Ref-L4 REC benchmark. We merge the validation and test set for evaluation.

| Model | mAcc | $Acc_{0.5}$ | $Acc_{0.75}$ | $Acc_{0.9}$ | mAcc-S | mAcc-M | mAcc-L |
|---|---|---|---|---|---|---|---|
| PixelLM 13B (Ren et al., 2023) | 44.3 | 67.1 | 48.2 | 16.7 | 20.4 | 47.8 | 55.6 |
| LISA (Lai et al., 2023) | 48.6 | 59.5 | 50.5 | 38.4 | 19.1 | 53.4 | 61.3 |
| PSALM (Zhang et al., 2024b) | 57.4 | 68.0 | 60.6 | 46.3 | 30.0 | 64.4 | 64.8 |
| GlaMM (Rasheed et al., 2023) | 55.2 | 66.1 | 57.9 | 44.9 | 20.2 | 63.6 | 65.6 |

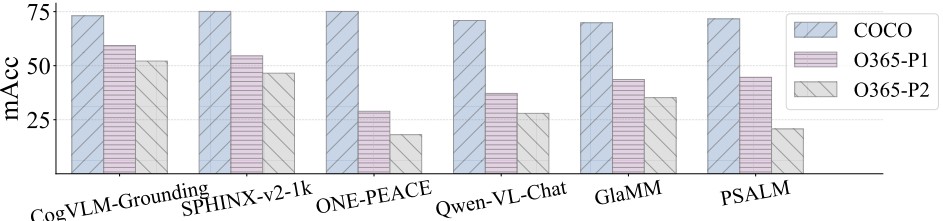

Figure 6: Evaluation of six models on various data sources, with mAcc acting as the metric. The results of all models can be found in Section C.2.

detailed in Table 6. Among the bounding-box-output models, CogVLM-Grounding (Wang et al., 2023b) excels with small and medium instances, while SPHINX-v2-1k (Lin et al., 2023) achieves the best performance with large instances. For mask-output models, GlaMM (Rasheed et al., 2023) outperforms all other models across all three sets.

**Evaluation on Diverse Data Sources.** Our benchmark is derived from COCO and Objects365 datasets. We assess the performance of the top four models with bounding box outputs and the top two models with mask outputs across various subsets originating from either COCO or Objects365. These subsets are: 1) the COCO-derived set (referred to as "COCO"); 2) a subset from Objects365, where the instances have categories that also exist in COCO (referred to as "O365-P1"); 3) another subset from Objects365, where the instances have categories not found in COCO (referred to as "O365-P2"). Figure 6 presents the performance of these models across the three subsets. The "COCO" set shows higher accuracy compared to the other two sets, partially because most models are trained on the RefCOCO series and have limited exposure to Objects365 images. "O365-P1" exhibits higher accuracy than "O365-P2", as the latter includes more rare categories.

**Extending to Referring Expression Segmentation.** The task of Referring Expression Comprehension (REC) can be extended to Referring Expression Segmentation (RES) by predicting a pixel-level mask instead of a bounding box. To extend our Ref-L4 for RES, we use a semi-automated process to transform the bounding boxes into mask annotations. Specifically, for each target instance and its corresponding image, we: 1) input the image and the target instance's bounding box into the SAM-2 (Ravi et al., 2024) model to generate an initial mask; and 2) manually review and correct the predicted mask if any inaccuracies are identified. We find that SAM-2's predictions are generally accurate, with only a small proportion of challenging cases (3.5%) requiring manual correction. Table 7 presents the evaluation of four models capable of predicting masks. The evaluation protocols remain consistent as above, except that the IoU is calculated between the predicted mask and the ground-truth mask. In each table, "S", "M" and "L" represent small, medium and large instances, respectively. In Figure 13, we provide visualizations of nine randomly selected segmentation annotations from our benchmark.

## 5 CONCLUSION

In this work, we first point out several limitations of the current REC benchmarks, such as substantial labeling inaccuracies and very brief referring expressions. To better assess the capabilities of models, particularly those LMMs that can perform the REC task, we present Ref-L4, which features four key characteristics: 1) a large-scale dataset with 45,341 annotations; 2) a wide range of object categories and varying instance scales; 3) detailed referring expressions; and 4) an extensive vocabulary comprising 22,813 unique words. We evaluate a total of 24 models using various evaluation protocols. We wish that Ref-L4 could serve as a valuable resource for researchers and developers, fostering the development of more robust and versatile REC models in the LMM era.

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

# APPENDIX

## A  LABELING ERRORS IN EXISTING BENCHMARKS

In the REC task, a referring expression should uniquely describe an instance, which is represented by an accurate bounding box. We have identified and visualized three common types of labeling errors in the RefCOCO, RefCOCO+, and RefCOCOg benchmarks: 1) non-unique referring expressions (Figure 7), which refer to multiple instances within the same image; 2) inaccurate bounding boxes (Figure 8); and 3) misalignment between target instances and their referring expressions (Figure 9), where the referring expressions are either ambiguous or do not refer to any instance in the image.

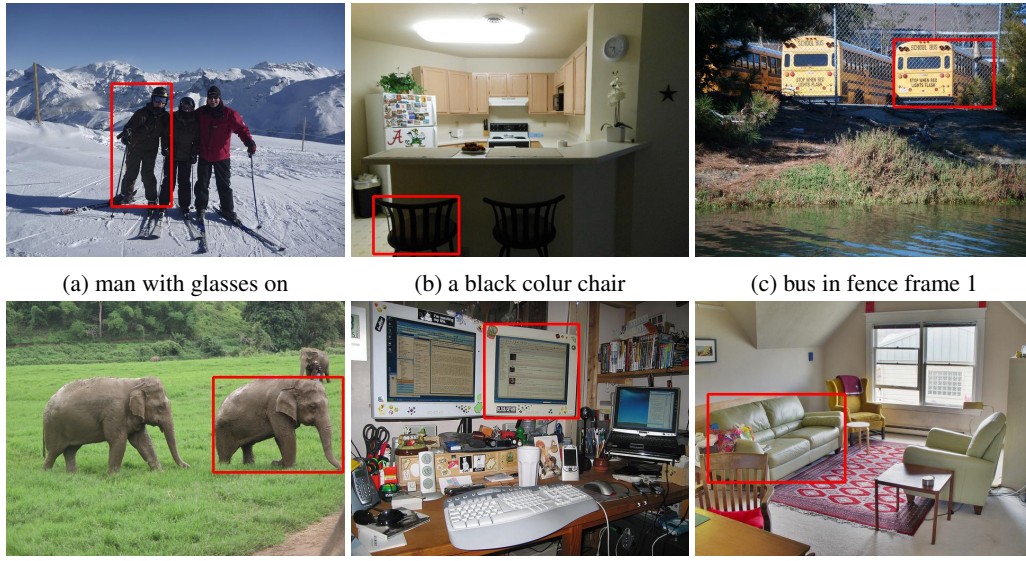

| | | |
|---|---|---|
| (a) man with glasses on | (b) a black colur chair | (c) bus in fence frame 1 |
| (d) an elephant walking in the grass | (e) a white computer screen | (f) white couch |

Figure 7: Visualization of labeling errors, where a referring expression refers to multiple instances within the same image. For each sub-figure, we display the original bounding box annotation with a red rectangle and include the corresponding referring expression in the caption.

## B  PROMPTS

### B.1  PROMPT FOR CONTEXT-INDEPENDENT DESCRIPTION GENERATION

Briefly describe the [*Category Name*] in one sentence. Begin your description with the object name, including adjectives if appropriate to describe its color or shape. Focus only on visible features and avoid mentioning blurriness.

Input image: [*Cropped Image*].

### B.2  PROMPT FOR CONTEXT-AWARE DESCRIPTION GENERATION

You are a sophisticated referring expression generator. Your task is to generate a clear and specific description for the target instance highlighted by a red circle in the provided image, based on a given hint and the following criteria:

*Criteria 1*: The description should enable individuals to understand and accurately identify the specified region within the image.

*Criteria 2*: The description may should various attributes such as category, shape, size, color, visibility, exposure, texture, orientation, absolute position, relative position, facial features, clothing,

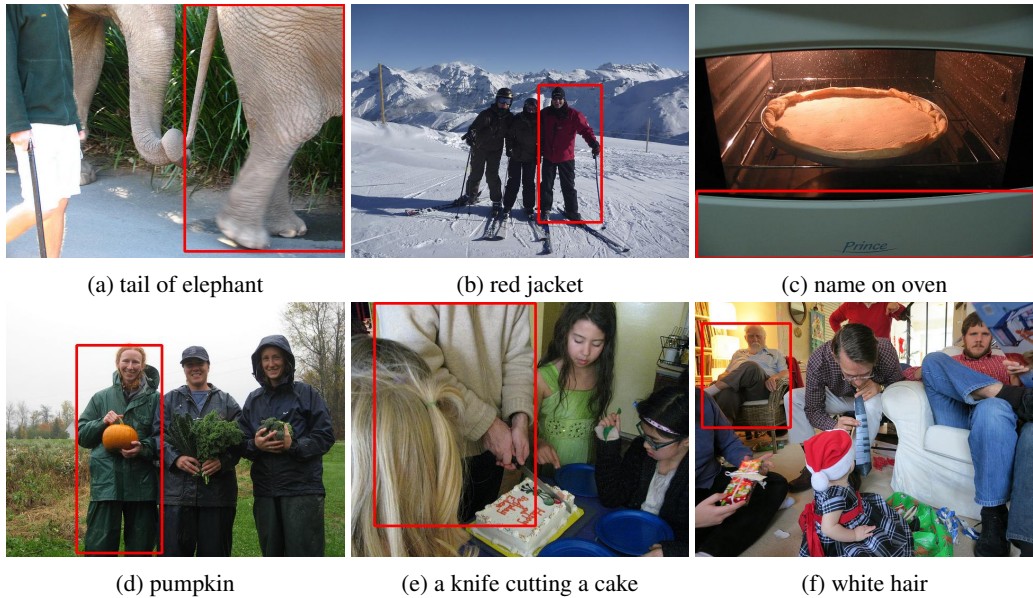

Figure 8: Visualization of labeling errors, where the bounding box annotations are inaccurate. For each sub-figure, we display the original bounding box annotation with a red rectangle and include the corresponding referring expression in the caption.

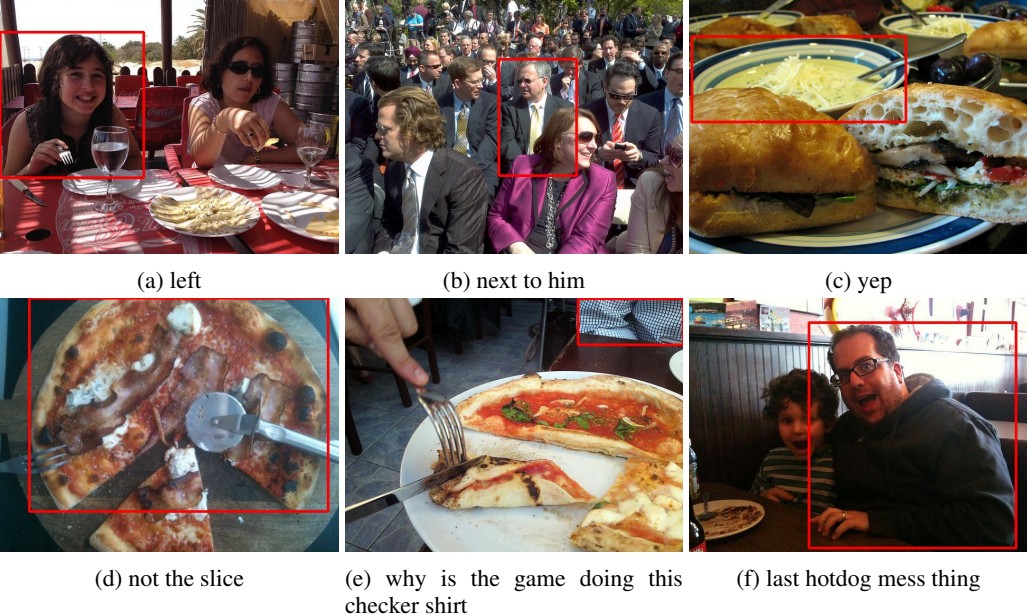

Figure 9: Visualization of labeling errors, where the referring expressions are either ambiguous or do not refer to any instance in the image. For each sub-figure, we display the original bounding box annotation with a red rectangle and include the corresponding referring expression in the caption.

accessories, gestures, context, semantic attributes, emotions, age, gender, posture, action, and especially interactions with other instances. The selection of features should be relevant to the particular region and the image context.

*Criteria 3*: The red circle is solely for highlighting the region of interest. Do not refer to it in your descriptions.

*Criteria 4*: Avoid using unnecessary words like "look for", "spot", "observe", "find", "notice", "identify", "outline", "target" and "question".

*Criteria 5*: Ensure that the subject of each sentence matches the subject given in the hints. Do not incorrectly use the subject as the object.

*Criteria 6*: Use the correct singular or plural form when referring to the target, which may be a single object, a pair of objects, or a group of objects.

*Criteria 7*: Integrate all relevant information from the hints, noting that some hints may be redundant or contain errors.

Input image: [*Raw Image*].

Hint: [*Context-Independent Description*].

### B.3 PROMPT FOR REPHRASING REFERRING EXPRESSIONS

Rewrite the subsequent description while preserving the main information. Utilize varied expressions and reorganize the sentences if necessary. Begin each sentence with the same subject being referred to.

Description: [*The Referring Expression to be Rephrased*].

### B.4 PROMPT FOR GPT4-V EVALUATION

You are an expert in referring expression comprehension and localization. Your task is to locate the object in the image based on the provided expression. The coordinates range from the top left $(0, 0)$ to the bottom right ([*Image Width*], [*Image Height*]). Please provide the bounding box in the format $(x_0, y_0, x_1, y_1)$, where $(x_0, y_0)$ represents the top-left corner and $(x_1, y_1)$ represents the bottom-right corner.

Expression: [*The Referring Expression*].

## C MORE EXPERIMENTS

### C.1 CATEGORY-WISE PERFORMANCE.

Figure 5 presents the per-category performance of the top four models. In Figures 10 and 11, we show the performance for all 24 models on a per-category basis, with mAcc serving as the metric, along with the average performance for each model across all categories.

### C.2 EVALUATION ON DIVERSE DATA SOURCES.

Figure 6 illustrates the performance of six models across three subsets, namely "COCO", "O365-P1" and "O365-P2". In Figure 12, the comprehensive results of 24 models across the same three subsets are displayed.

## D LIMITATIONS AND BROAD IMPACTS

Ref-L4 provides a more comprehensive and detailed evaluation of REC capabilities, helping to better understand and improve the performance of large multimodal models capable of handling the REC task. The public availability of Ref-L4 and its evaluation code encourages further research and collaboration, driving innovation and advancements in the field of REC and beyond. While Ref-L4 aims to cover a wide range of scenarios, it may still miss out on specific edge cases or unique contexts that could be encountered in real-world applications. The detailed and lengthy referring expressions might pose a challenge for current models, requiring significant advancements in natural language processing and comprehension capabilities.

# E   AUTHOR STATEMENT

The authors of the Ref-L4 benchmark accept full accountability for any rights violations, such as copyright infringement or other legal breaches. They emphasize that all data included in the Ref-L4 dataset adheres to the licensing agreements of the original source datasets. The Ref-L4 benchmark is made available under the Creative Commons Attribution-NonCommercial 4.0 International (CC BY-NC 4.0) license. Meticulous attention has been paid to ensure that the dataset upholds the highest legal and ethical standards. The authors are committed to addressing any issues arising from the use of this dataset and stand prepared to take necessary actions to resolve them.

# F   MAINTENANCE AND LONG TERM PRESERVATION

To ensure the benchmark remains relevant and useful for evaluating REC models, we will establish a protocol for regular updates. This includes the addition of new image sets and text annotations that reflect current trends and challenges in the field. A version control system will be implemented to track changes and updates to the benchmark. Each version will be documented with detailed notes on the modifications, including the addition of new data, changes to annotation guidelines, and improvements based on user feedback. We will utilize reliable cloud storage solutions with multiple redundancy mechanisms to safeguard against data loss.

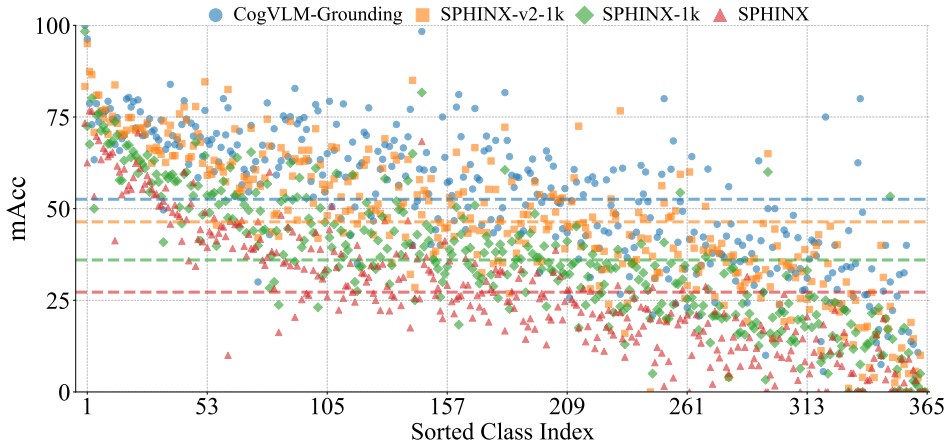

(a) The average performance across all categories (dot lines) for CogVLM-Grounding Wang et al. (2023b), SPHINX-v2-1k Lin et al. (2023), SPHINX-1k Lin et al. (2023), and SPHINX1 Lin et al. (2023) are 52.56, 46.40, 36.01, and 26.95, respectively.

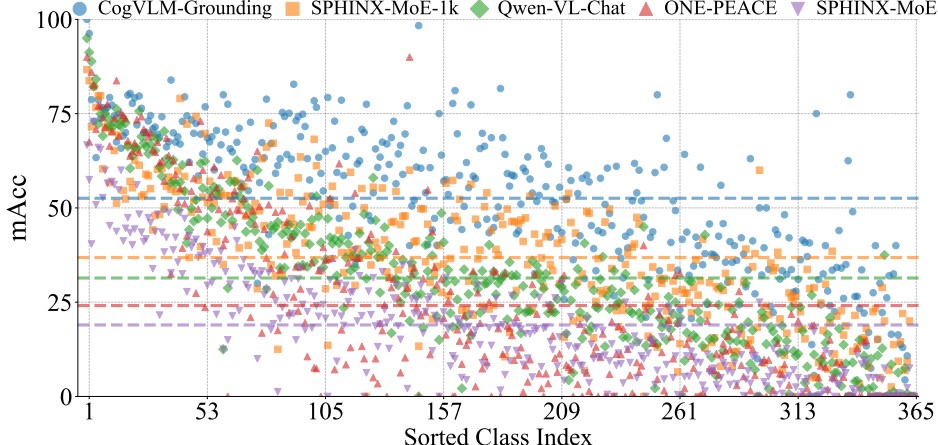

(b) The average performance across all categories (dot lines) for SPHINX-MoE-1k Gao et al. (2024), Qwen-VL-Chat Bai et al. (2023), ONE-PEACE Wang et al. (2023a), and SPHINX-MoE Gao et al. (2024) are 36.84, 31.41, 24.11, and 18.77, respectively.

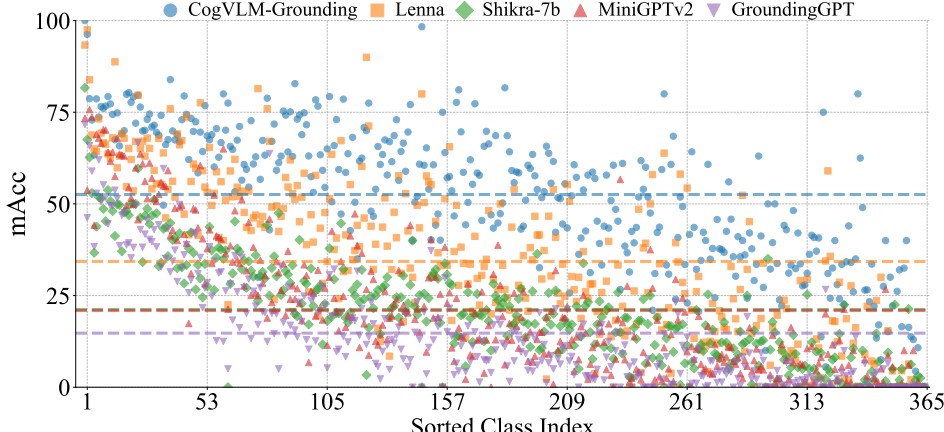

(c) The average performance across all categories (dot lines) for Lenna Wei et al. (2023), Shikra-7b Chen et al. (2023b), MiniGPTv2 Chen et al. (2023a), and GroundingGPT Li et al. (2024) are 34.30, 21.22, 21.13, and 14.60, respectively.

Figure 10: Category-wise performance of 24 models (part-1), sorted in the same order as in Figure 5. We use CogVLM-Grounding as a reference for comparison in each sub-figure.

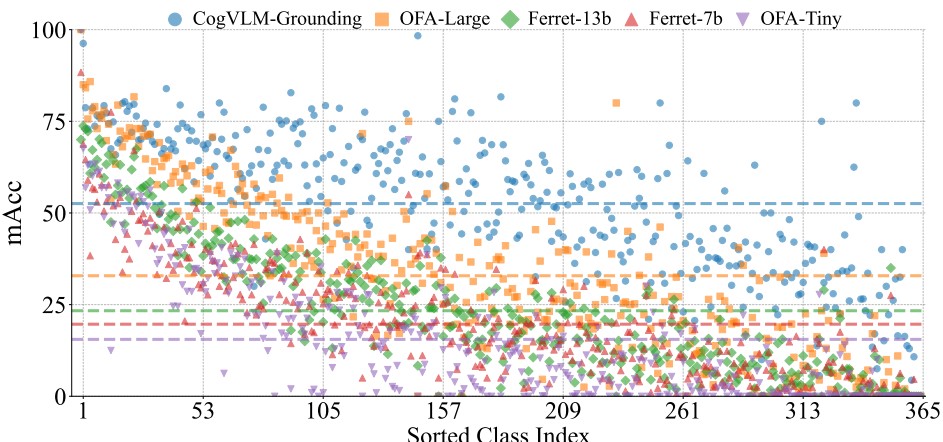

(a) The average performance across all categories (dot lines) for OFA-Large Wang et al. (2022), Ferret-13b You et al. (2023), Ferret-7b You et al. (2023) and OFA-Tiny Wang et al. (2022) are 32.88, 23.33, 20.27, and 15.37, respectively.

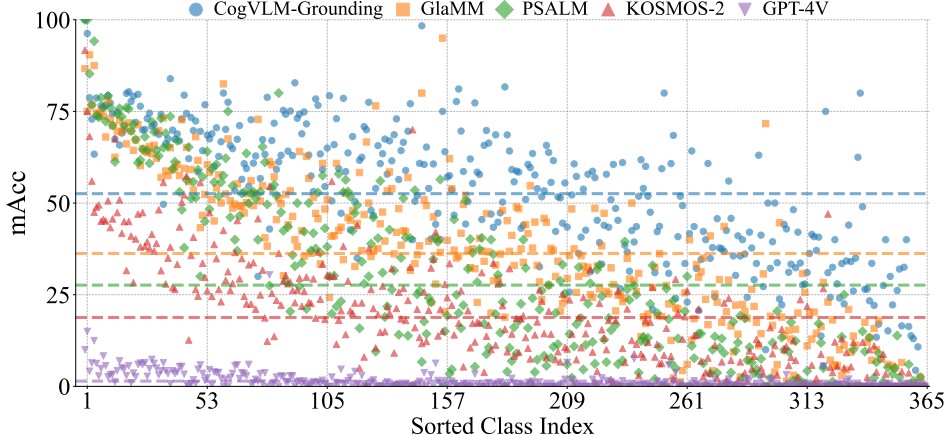

(b) The average performance across all categories (dot lines) for GlaMM Rasheed et al. (2023), PSALM Zhang et al. (2024b), KOSMOS-2 Peng et al. (2023) and GPT-4V OpenAI (2023a;b;c) are 36.25, 27.62, 19.37, and 1.42, respectively.

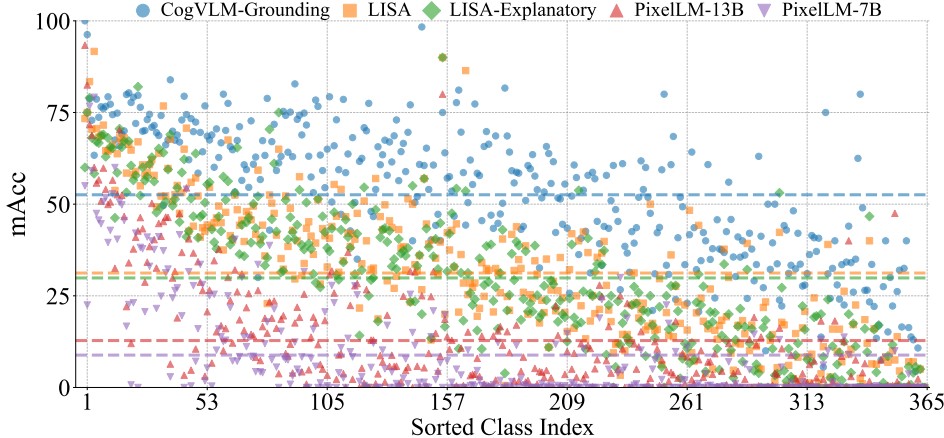

(c) The average performance across all categories (dot lines) for LISA Lai et al. (2023), LISA-Explanatory Lai et al. (2023), PixelLM-13B Ren et al. (2023) and PixelLM-7B Ren et al. (2023) are 31.22, 29.87, 13.19, and 8.74, respectively.

Figure 11: Category-wise performance of 24 models (part-2), sorted in the same order as in Figure 5. We use CogVLM-Grounding as a reference for comparison in each sub-figure.

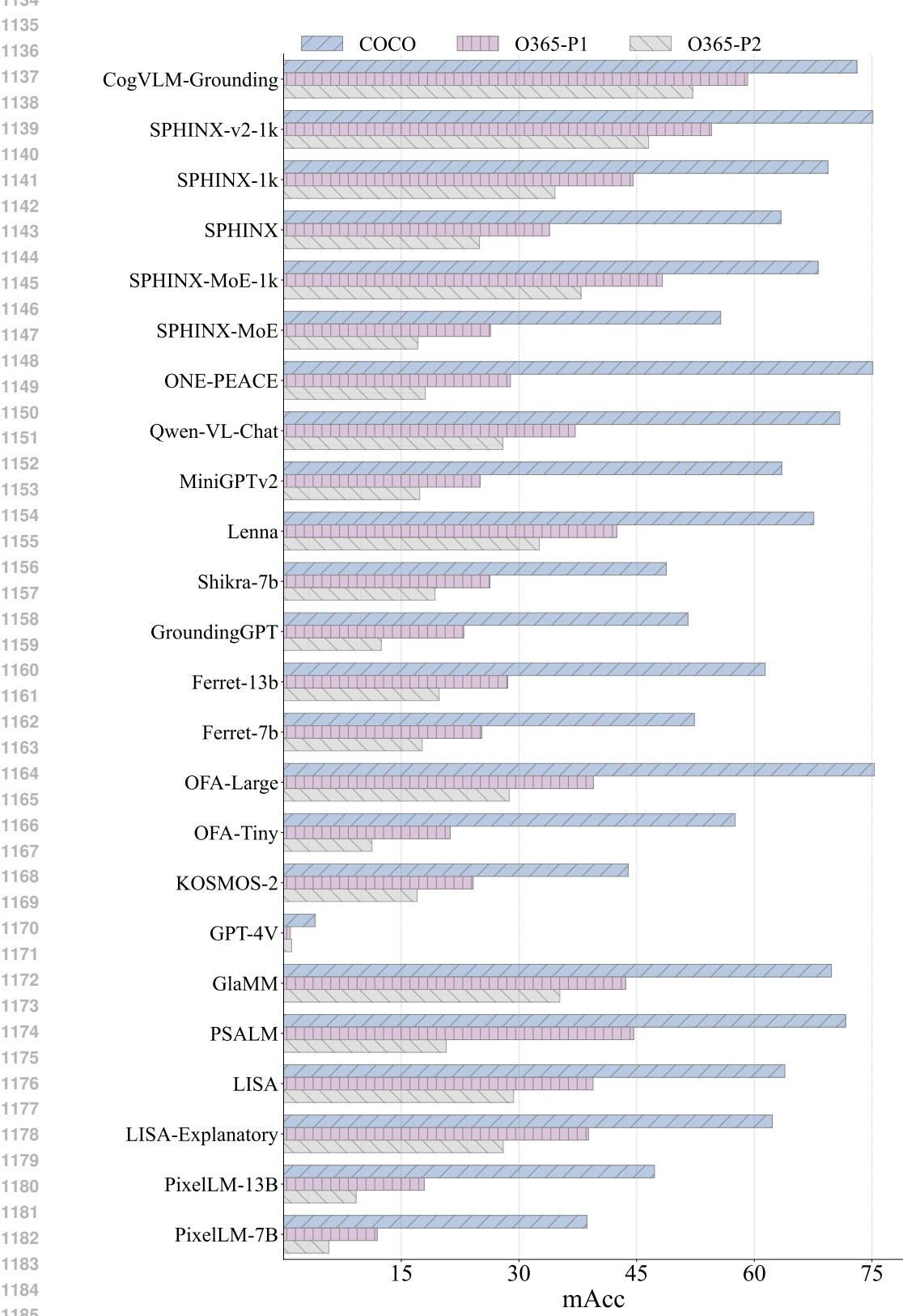

Figure 12: Evaluation of 24 models on various data sources, with mAcc acting as the metric.

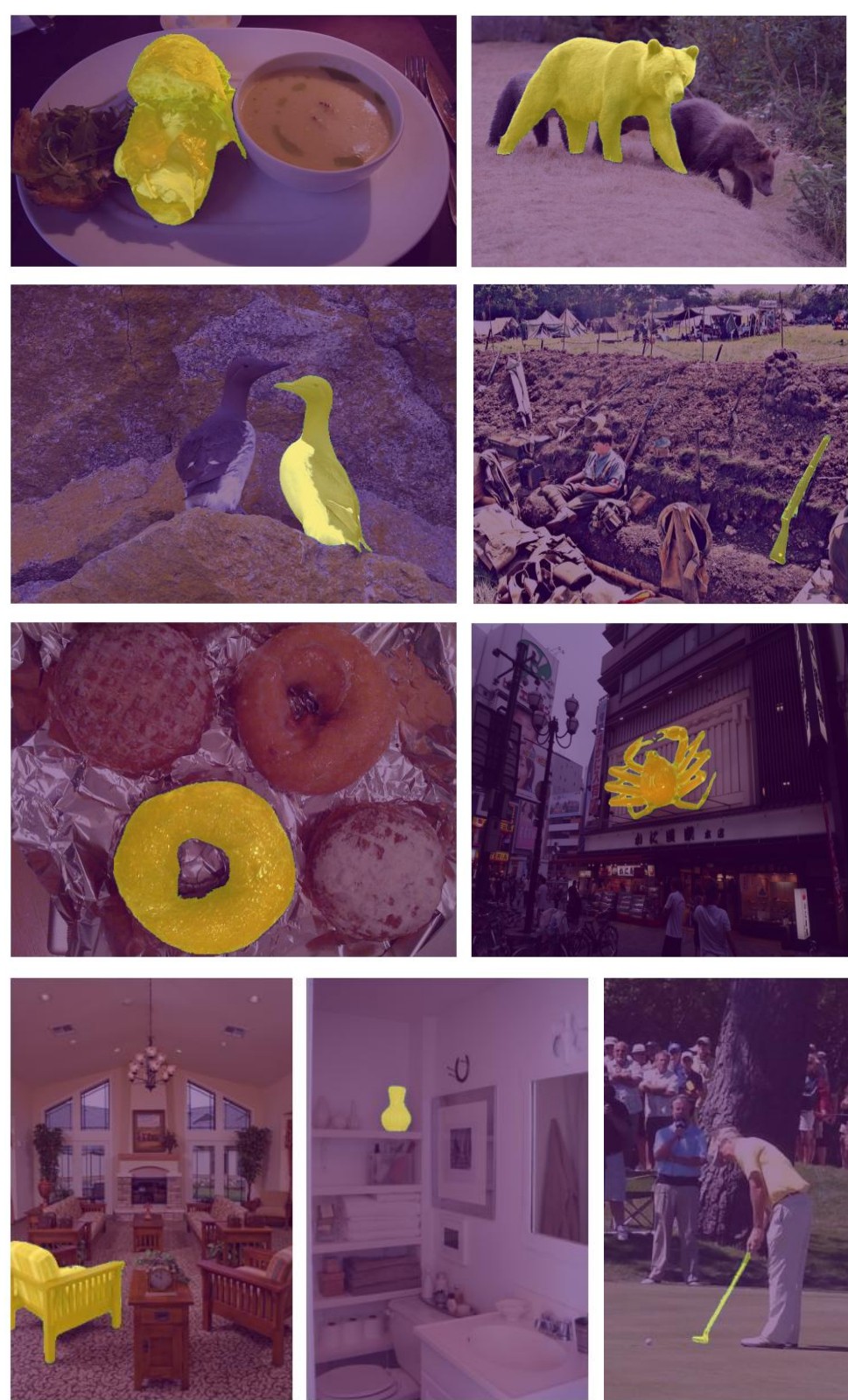

Figure 13: We provide visualizations of nine randomly selected segmentation annotations from various categories within our benchmark. The annotations are highlighted in yellow.

