# OpenReview forum: "Revisiting Referring Expression Comprehension Evaluation in the Era of Large Multimodal Models"
_ICLR.cc/2025/Conference — ICLR 2025 Conference Withdrawn Submission_

### Official Review · Reviewer_rSpB · 2024-11-02

**Soundness:** 3
**Presentation:** 2
**Contribution:** 3
**Rating:** 5
**Confidence:** 3

**Summary:**

This paper presents the Ref-L4 dataset, generated using GPT-4V, which features a substantial number of test samples, extensive diversity across object categories and instance scales, long referring expressions, and a broad vocabulary. By constructing a comprehensive benchmark, they compare the performance of multiple methods, addressing the gap in available datasets with sufficiently detailed referring expressions.

**Strengths:**

They constructed a comprehensive benchmark, comparing results across various methods. This paper effectively addresses the issue of insufficient referring expressions and makes efforts toward resolving it.

**Weaknesses:**

- It is recommended to include more examples to illustrate the referring expressions of REF-L4, as the current version provides relatively few samples. Additionally, in Figure 1, the expression “It is placed on the floor” appears to be somewhat inaccurate. The current experiments seem insufficient to demonstrate that the dataset descriptions are entirely accurate, and there is limited information on the manual review process. Providing additional examples would help readers gain a fuller understanding of the dataset.
﻿
- Is a longer “referring expression” necessarily better? It would be helpful if the authors could discuss this point. Given that the GPT-4V model generates lengthy descriptions to comprehensively cover object details, such extended expressions may not align with real-world applications, where shorter descriptions are often preferred. There is some concern that this dataset may not fully reflect model performance when working with shorter descriptions.

**Questions:**

- Could you provide more cases?
Additional examples would help illustrate the typical structure and content of the referring expressions in this dataset. This would offer readers a clearer picture of what the dataset includes and its potential applications.
- Could you include a description of the manual review process?
A detailed explanation of the manual review process would enhance the dataset’s reliability. Understanding how the data was verified and any measures taken to ensure quality would give readers more confidence in the dataset’s accuracy.
- Could you discuss the impact of expression length?
It would be beneficial to address whether longer referring expressions are preferable. While the GPT-4V model generates detailed descriptions to thoroughly cover object attributes, shorter expressions are often more practical in real-world applications. A discussion on how expression length affects model performance and use-case relevance would provide valuable insights for readers and users of the dataset.

---

### Official Review · Reviewer_XSZk · 2024-11-05

**Soundness:** 2
**Presentation:** 3
**Contribution:** 2
**Rating:** 3
**Confidence:** 4

**Summary:**

This paper focuses on the evaluation benchmarks of REC models
It firstly notices some problems with existing REC benchmarks, and discuss them detailedly.
Then, it cleans the popular refcoco datasets and build a ref-l4 benchmark upon them.
The proposed ref-l4 benchmark is larger, cleaner, with longer descriptions and larger vocabulary.

**Strengths:**

- The paper is generally well-written. The contributions are clearly stated and understandable. The authors provide qualitative cases and quantitative comparisons (like Tab.3) to help understand the major differences between this work and existing works.
- The evaluation of many LMMs on the proposed benchmark can be useful for reference. This also requires some experimental efforts. It is appreciated.

**Weaknesses:**

The weaknesses, from the most significant to the least significant:

- The contribution of the dataset seems incremental. This is my biggest concern. This work majorly presents a better version of RefCOCO datasets for evaluating REC models. The major differences includes larger dataset size, vocabulary size and longer descriptions. These improvements are good, but not essentially significant. The task form is the same; the scale is basically twice of previous works (for number of images, instances, etc.), which is not significant enough to me.
- The relationship between this work and LMMs are not strong enough. The core contribution is a dataset for REC models, with better quality and (not much) larger size. This is not strongly related to LMMs. The only significant relationship between this work and LMMs is the evaluation in Sec. 4. However, this is only an evaluation with experimental results listed and the authors did not provide much insight about LMMs.
- The title "in the era of LMMs" is surprising but the actual contribution is not satisfying. As discussed right above, the core contribution of a REC benchmark with better quality and (not much) larger size, is not very connected with "the era of LMMs". The evaluation with LMMs does not provide much information about how we should tackle REC "in the era of LMMs". I would admit that I am a little disappointed by the contrast between the title and the actual contribution.
- Some related works are not discussed, e.g., Omnilabel (ICCV 2023) and Described Object Detection (NeurIPS 2023). They also propose evaluation benchmarks with more instances and longer expressions. If these benchmarks are considered, then the major differences between this benchmark and previous ones, like scale and description lengths, seem to be invalid. The authors may consider discuss the comparisons with them.

**Questions:**

Currently I lean towards the recommendation of rejection. Please see the weaknesses above.

It would helps if the authors can explain more on the contribution of this dataset (besides better labeling and twice larger scale) compared with REC benchmarks.

Also, I would appreciate it if the authors can either (1) clarify the major relationships between this benchmark and LMMs, or (2) consider reducing the referring to LMMs if their differences with traditional grounding models are not truly essential to this work.

---

### Official Review · Reviewer_YvBc · 2024-11-06

**Soundness:** 2
**Presentation:** 3
**Contribution:** 2
**Rating:** 3
**Confidence:** 5

**Summary:**

This paper aims to address the task of referring expression comprehension. Authors argue that there are some error cases in previous popular datasets, and previous datasets are not enough to  test the LLM’s REC performance. From this point, authors proposed a LLM generated dataset, and evaluated 24 LLMs on the proposed dataset.

**Strengths:**

1. The paper proposes a large-scale referring expression dataset and an LLM-based data generated pipeline.
2. The paper also identifies some error cases in previous datasets.
3. A large number of methods are collated and evaluated in this paper, which is very impressive.

**Weaknesses:**

1. The proposed dataset is still a traditional referring dataset. Since there are many advanced referring expression datasets are available, e.g. generalized referring segmentation dataset gRefCOCO, reasoning dataset LISA, sub-object dataset PhraseCut, the value of proposing a new traditional referring expression dataset is relatively limited.
2. Some statistics of the dataset are questionable, e.g.
    2.1. The expansion in category count largely stems from the base dataset Object 365, rather than as an inherent contribution of the proposed dataset.
    2.2. It appears that “instance area” is calculated in absolute pixel numbers, which may be less informative than the relative area of object pixels compared to the entire image, as the latter provides more practical insight.
3. Author stated that there exist a lot of error cases in test and validation set of previous datasets - why only focus on test and val set, how about training set? And what is your verification process? Table 2 is also confusing - from my understanding, the conclusion of this table is: removing wrong samples from testing set can increase the numeric performance of methods - which could be too obvious and expected.
4. From Introduction, the main motivation of the paper is to “*existing REC benchmarks truly capture the comprehensive capabilities of LMMs*”, and the solution is to proposed a dataset. However, in my opinion, all of the four main features of the new dataset (L134-L150) are not exclusive to LLMs but can also be well addressed by non-LLM methods. Authors should give more justifications about why these features can help your dataset to “*capture the comprehensive capabilities of LMMs*” but previous datasets cannot.
5. For example, longer expressions do not necessarily mean harder cases. Actually, as author instructed LLM to “*describe various features such as color, size, position, and context*”, the generated expressions may tend to contain more information and more details, which in contrast could make them easier.
    5.1. Additionally, longer expressions may lack practical relevance. In real-world scenarios, people typically use brief, direct language to refer to objects rather than lengthy descriptions.
6. What is the process of “*manually review*” of the dataset? Since most of the expressions are generated by LLM, more details are required here to ensure the validity of the annotations.

**Questions:**

I am impressed by the authors’ effort to compile, adapt, and evaluate a substantial number of LLM-based methods for referring expression comprehension. However, the motivation behind the proposed dataset does not feel fully convincing. Some design considerations are unclear, and as a traditional referring expression dataset, it appears somewhat conservative, offering limited new insights to the field.

---

### Note · Authors · 2024-11-12

**Comment:**

We thank the reviewers for their time and feedback on our paper. After careful consideration, we believe our work may not be the best fit for ICLR at this time, and we have decided to withdraw our submission.

**Withdrawal Confirmation:**

I have read and agree with the venue's withdrawal policy on behalf of myself and my co-authors.